# Towards Robust Exocentric Mobile Robot Tele-Operation in Mixed Reality

Ke Li * †, Reinhard Bacher*, Wim Leemans*, Frank Steinicke †

*Deutsches Elektronen-Synchrotron DESY* *, Hamburg, Germany

Human-Computer Interaction, *Universität Hamburg* †, Hamburg, Germany

*Abstract*—A mobile robot is an important type of robot commonly used to explore remote environments that are hazardous and inaccessible for humans. However, operating a mobile robot to explore a remote environment using the conventional 2D user interfaces could be challenging, especially when exploring an environment where unexpected robot behavior has to be avoided. In recent years, using mixed reality technology for human-robot interaction has demonstrated benefits in increasing robot operation safety, improving human trust in robots, and reducing operator's mental and physical workload.

In this work, we describe a novel tele-operation system that could be used for mobile robot navigation and manipulation in a human-robot co-located mixed reality environment or a digital twin of the remote environment, where users could freely switch perspectives in the mixed reality environment. We first present our interaction design and system implementation and then discuss the future work for user studies and evaluations.

*Index Terms*—Mixed Reality, Virtual Reality, Human-Robot Interaction, Mobile Robot, Robotics Control

## I. INTRODUCTION

A mobile robot is an important type of robot, which assists humans in exploring inaccessible remote environments. Although, advanced sensory technology and intelligent system algorithms have made mobile robots become more and more autonomous [1], in hazardous and dangerous environments such as particle accelerator tunnels and nuclear facilities, where unexpected robot behaviors have to be avoided, mobile robot operation and manipulation still largely depends on manual or semi-autonomous tele-operation [18]. Therefore, developing effective and easy-to-use operation and manipulation methods for a remote mobile robot is an important research topic in the field of human-robot interaction (HRI).

In recent years, the rapid development of mixed reality (MR), augmented reality (AR), and virtual reality (VR) technology (together abbreviated here as VAM) has become a new trend for HRI. The availability of high-quality VAM head-mounted displays (HMD) has shifted mobile robot operation from the conventional 2D user interfaces (UI) on a computer screen [11], [12] to an immersive 3D UI on MR HMD [4], [10]. Past research has demonstrated that users, in general, favored MR HMD over the conventional 2D display for mobile robot operation, as MR UI enables a more natural and immersive tele-presence system with enhanced situation awareness and sense of presence [20]. However, previous MR tele-presence systems based on video streaming of stereoscopic or 360 cameras could only provide an egocentric viewpoint [2]. Such egocentric constraint makes mobile robot operations

such as robot path planning for long-distance traveling and real-time dynamic path following in a narrow and non-robot-friendly environment challenging.

In this work, we extend previous research in using MR for mobile robot operation. We present a system and several interactions designs based on an intuitive MR HMD user interface for performing tasks such as robot path planning from an exocentric perspective. Our system and designs target applications where users could freely navigate inside the virtual environment, such as a human-robot co-located MR video-see-through (VST) or optical see-through (OST) environment, or inside a structured virtual digital twin. In addition, we implement our system based on an open-source software stack and the Anki Cozmo education robot [1]. The implementation code for the proposed system and designs will be made publicly available, and could be easily set up and experimented with for future research and studies [2].

In this extended abstract, we first provide an overview of the system architecture. Then, we describe the implementation and design of a path following algorithm. Moreover, we present two interaction techniques for exocentric robot path planning in MR. Finally, we discuss a specific use case of our system and design, the current limitation of the system, and future plans for evaluations and user studies.

## II. PROTOTYPE DESIGN

### A. System Setup

Figure 1 illustrates the basic components and structures of our MR mobile robot interaction system. The client side of the system is an application built with the popular game engine Unity, which can be run on any MR device that supports the OpenXR backend [8]. The MR client can be an OST HMD such as the Microsoft Hololens2 [3], or a VST HMD such as the Oculus Quest2 VR headset with passthrough API [13]. Using OpenXR backend allows our system to be compatible with a wide range of HMDs. Additionally, we use the MRTK, a popular MR development framework for creating immersive user interfaces.

Our system targets Anki Cozmo is an education mobile robot that is easy to set up and develop. The Cozmo robot has four differential wheels and a movable lift. The Cozmo

---

[1]https://www.digitaldreamlabs.com/pages/cozmo
[2]The repository is available here https://github.com/keli95566/VAMCozmo
[3]https://www.microsoft.com/en-us/hololens/

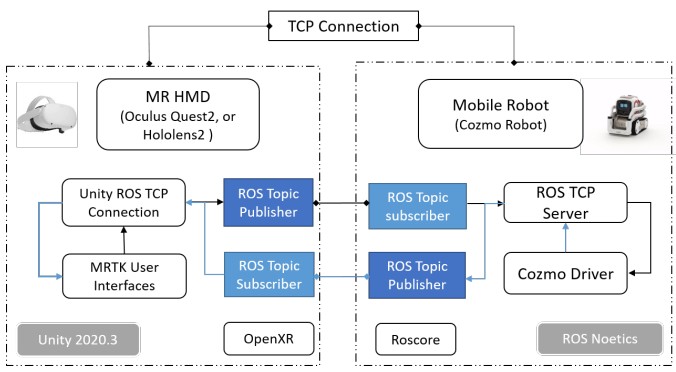

Fig. 1. Illustration of the system overview: The system consists of an MR HMD client running on Open XR backend, and a Cozmo robot running on ROS. The robot and client are connected via TCP connection over a local network.

Driver [14] is a Python package that connects the Cozmo SDK with the robot operation system (ROS) to send commands and receive sensory feedbacks from the robot. We establish communication between the MR HMD client and the robot via the Transmission Control Protocol (TCP) connection over local network using the ROS-TCP connection package provided by Unity [19].

The proposed system creates a simple and easy-to-scale design space for developing HRI techniques in MR and could be used for both VST HMD and OST HMD. Figure 2 illustrates an interaction example for the Cozmo robot and Hololens 2 using the proposed system. In the example, users could control the movable lift of the robot via hand tracking in real-time. A line is rendered between the index and thumb finger to measure and control the height of the movable lift.

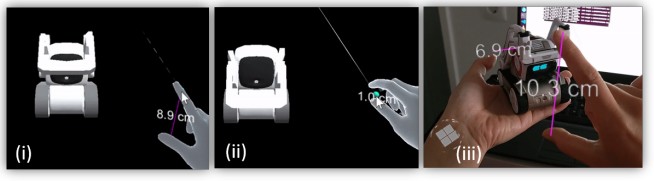

Fig. 2. An example interaction with the Cozmo robot in MR with our system. Sub-figure (i) and (ii) illustrate our interaction designs in simulation. Sub-figure (iii) illustrates live control of the robot lift using our system with Microsoft Hololens 2.

### B. Differential Drive Robot Kinematics

Cozmo is a simple differential wheel drive robot. The movement and direction change of the robot depends on the relative rate of rotation of wheels on either side of the robot body and does not require additional steering. To correctly simulate a mobile robot's locomotion behavior in MR, we implement an articulation wheel controller based on the classical kinematics model for differential drive robots [9]. In contrast to a typical two differential wheel drive robot, the Cozmo robot is a four-wheel differential drive robot. Therefore, to apply the classical kinematics model, we make the following assumptions:

1) The robot is moving on a 2D plane with constant friction. Therefore, the robot locomotion state could be expressed by a state vector $R = (x, y, \theta)$, where $x$ and $y$ are the 2D positions of the robot, and $\theta$ is the angle that describes the direction the robot is facing.
2) The two wheels on the same sides drive at the same linear speed. Therefore, we could use the same kinematic model for a robot with two differential wheels for the four differential wheel Cozmo robot.

Given a target linear speed $v(m/s)$ and angular speed $\omega(rad/s)$, the expected linear speed of the right wheels ($v_r$) and the speed of the left wheels ($v_l$) of the robot could be calculated using the following equation:

$$v_r = \frac{l \cdot \omega}{2} + v \tag{1}$$

$$v_l = -\frac{l \cdot \omega}{2} + v \tag{2}$$

where $l$ is the distance between the left wheel and the right wheel.

With the linear and angular velocity, we could calculate the joint speed of individual differential wheel $\omega_i = \frac{v_i}{R_i}$, where $v_i$ is the target linear speed of the wheel, and $R_i$ is the radius of the wheel.

### C. Path-following Algorithm

Based on the differential drive kinematics model, we develop a simple path following algorithm which allows robots to follow a pre-planned path defined through a list of way-points. As described in Algorithm 1, the path following algorithm utilizes the simplest forward kinematics commands such as turn in place and drive straight, and does not depend on an inverse kinematics calculation [16] which requires more constraint consideration.

---

**Algorithm 1** A Simple Path Following Algorithm for Differential Drive Robot

---

**Input:** A list of way-points $P$ that define the path for robot to follow, and robot's initial state $R_0 = (x_0, y_0, \theta_0)$.
**Output:** Robot trajectory
1: **for** each point $p(x, y)$ in $P$ **do**
2:     **while** Robot not pointing towards the target point $p(x, y)$ **do**
2:         Turn robot in place with target angular velocity $\omega$.
2:         Update robot state vector $(x_i, y_i, \theta_i)$.
3:     **end while**
3:     Calculate distance $d$ between robot's current position $(x_i, y_i)$ and the target point $p = (x, y)$.
4:     **while** Robot not traveling sufficient distance $d$ **do**
4:         Drive robot forwards in the target direction with the target linear velocity $v$.
4:         Update robot state vector $(x_i, y_i, \theta_i)$.
5:     **end while**
6: **end for**

---

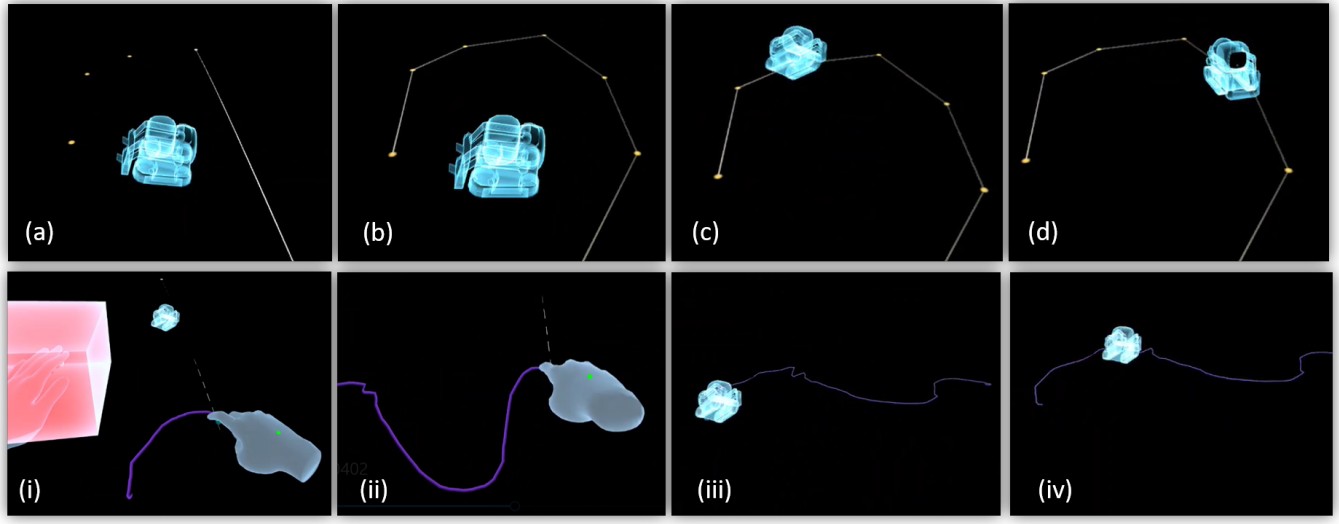

Fig. 3. Illustration of two different MR mobile robot path planning interaction designs: Sub-figure (a-d) illustrate a raycast pointing method, where users define and select way-points via hand or controller pointing. Sub-figure (i-iv) illustrates a free-hand drawing method, where the drawn trajectory is projected onto a 2D plane as the robot's final motion trajectory.

## D. Interaction Designs

One major advantage of an exocentric MR environment for mobile robot operation is the possibility to accurately plan the robot's future trajectory inside the entire virtual or MR environment. Figure 3 illustrates two interaction designs for trajectory planning of the mobile robot.

In the first design, the user can define the robot's future path via controller or hand pointing. An arrow-like pointer originates from the user's hand or controller intersects with a point on the 2D plenary surface, thus providing accurate visual feedback of the pointed position. After selecting multiple target points in the 3D environment (Figure 3 a), the user could confirm the path (Figure 3 b), and the robot will sequentially move to different way-points selected by the users (Figure 3 c-d).

In the second design, the robot follows a trajectory that the user defines via free-hand drawing. The drawing begins with the user touching a trigger button (Figure 3 i). A trajectory is created via tracking the 3D position of the index finger of the user's hand (Figure 3 ii), and the drawing terminate with the user's other hand leaving the trigger button. Further, the initial list of the tracked 3D points is filtered, sampled, and optimized, before being projected onto the 2D floor. The robot then runs the path following algorithm 1 (Figure 3 iii-iv) by reaching different way-points on the drawn trajectory.

Past research has shown that controller or hand pointing is the most effective selection method for selection task that requires accurate pointing, such as VR text selection and typing [17]. Due to a lack of visual feedback, freehand drawing could be less accurate and more confusing. However, it could offer a quicker and more intuitive way for the operator to draft and illustrate the general robot trajectory [15]. Although for accurate positioning of the robot, the controller and hand

pointing method could be more practical, the freehand drawing method could be useful when it comes to path illustration or robot path drafting on a world in miniature (WIM) representation or a 2D map of the environment. Future user studies will evaluate and validate the system usability [3] and workload index [7] for each of the proposed interaction designs.

## III. A Real-World Use Case

One important application of the proposed system and design is robot trajectory and motion planning at a large-scale physics facility such as the linear accelerator (LINAC) tunnel for the European X-Ray Free-Electron Laser facility (EuXFEL) [4]. As particle accelerator tunnel has a highly radioactive environment during run time, using mobile robots is a highly attractive solution for facility inspection and maintenance [5], [18]. Additionally, the accelerator tunnel is a narrow and robot non-friendly environment. Using MR user interfaces for mobile robot tele-operation could greatly enhance the operator's situation awareness, sense of presence, and level of trust in the remote robot. However, a particle accelerator is a large-scale facility, using the conventional egocentric MR tele-presence system could not facilitate task planning that involves large space and long-distance.

Luckily, the general 3D structure of the facility already exists in the digital twin or the CAD model. Therefore, an exocentric approach towards robot task planning in MR has the potential to improve operational accuracy and efficiency. Figure 4 illustrate an example scenario of tele-operation of a mobile robot at the EuXFEL LINAC facility using an exocentric MR user interface where our proposed system and interaction designs could be applied to.

[4]https://www.xfel.eu/facility/accelerator/index_eng.html

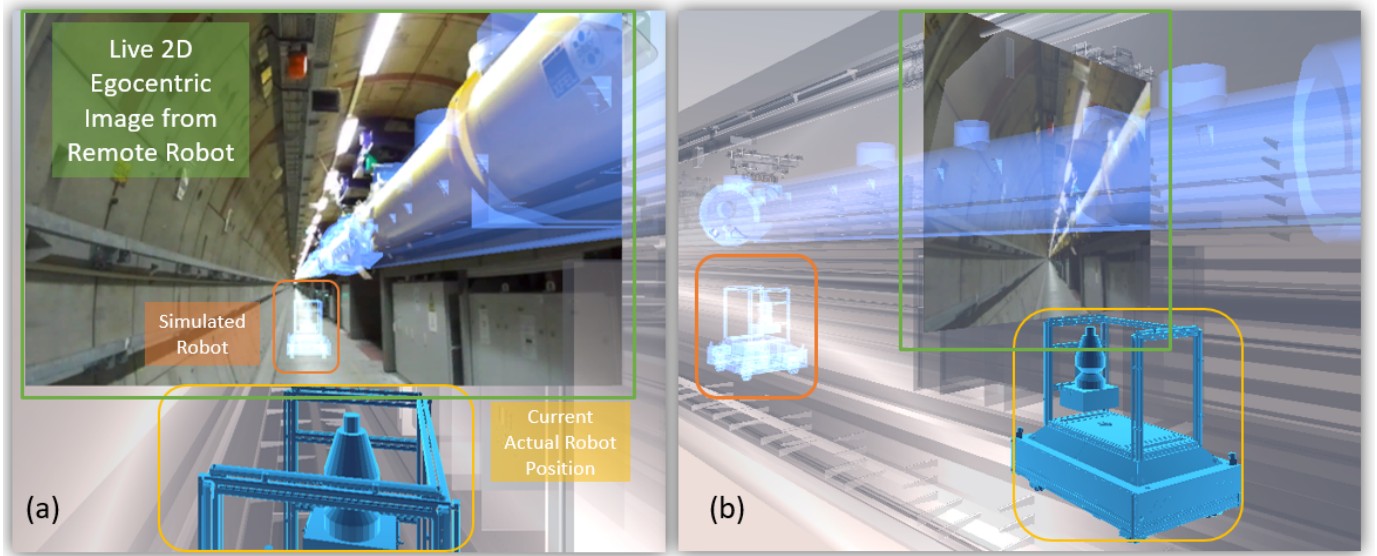

Fig. 4. Example scenario of tele-operation of a mobile robot at the EuXFEL LINAC facility using exocentric MR user interfaces with the support of 2D egocentric video lives stream monitoring.

On one hand, robot operators could monitor the actual remote environment via an egocentric 2D video live stream sent by the remote robot to monitor unexpected changes in the environment. As illustrated in Figure 4 a, the video live stream is rendered as an additional layer and merged into the virtual environment without completely breaking the operator's sense of presence. As 2D video live stream does not depend on large network bandwidth, mixing a 2D live stream with the 3D virtual environment could overcome the issue of network latency of previous tele-presence system based on 360 cameras.

On the other hand, operators could also switch to any exocentric perspective for investigating the virtual environment and planning future action, as illustrated in Figure 4 b. Additionally, a time delay is maintained between the simulated virtual robot in case anything unusual occurs in the simulation, the operator has time to halt the operation of the actual robot, creating an additional safety layer for robot tele-operation in hazardous environments.

## IV. FUTURE WORK

One important future step for this work is to evaluate the proposed system and investigate various human factors of the proposed designs when applied to the suggested scenario for mobile robot operation at the EuXFEL accelerator. Within-subject studies will be carried out to compare the usability and task completion efficiency of different interaction schemes for mobile robot path planning inside a large-scale physics facility such as a particle accelerator tunnel. We will compare the proposed interaction methods with the conventional 2D user interfaces, and a strictly egocentric tele-presence robot based on VR [2] or 360 cameras [21]. We plan on measuring task completion time and success rate, the system usability scale (SUS) [3], operation workload index (NASA-TLX) [7], and

the level of operator's trust in the remote robot [6]. Additionally, expert feedback from the mobile robot operators at the EuXFEL will be collected via a semi-structured user interview to evaluate the usability, perceived safety, advantages, and limitation of the proposed system and design.

In addition, our system and design could be further expanded to planning the 3D movement of a robot arm along a pre-defined path in a narrow, complex, and non-robot friendly environment.

One other improvement we aim for in the future is to match robot's simulated kinematics behavior in MR to the real world. The current simulated kinematics model makes assumptions that neglect several important real-world conditions such as friction, skid factor, and wheel slip, which could alter the robot's real trajectory from the simulated path. Implementation and testing of an advanced kinematics model will improve the performance of our system and designs.

## V. CONCLUSION

In this work, we propose a system and several interaction designs for exocentric robot operation in MR. We implement a simple articulation wheel controller and path-following algorithm for an education robot as an example and present two MR interaction designs for robot trajectory planning. Finally, we demonstrate that our proposed system and inter-action designs have the potential to allow better HRI in a structured virtual remote environment or a human-robot co-located environment such as a particle accelerator facility. Our future work will focus on evaluating the proposed system and investigating various human factors of the proposed designs.

## ACKNOWLEDGMENT

This work was supported by DASHH (Data Science in Hamburg - HELMHOLTZ Graduate School for the Structure

of Matter) with the Grant-No. HIDSS-0002. In addition, we thank Andre Dehne, Julien Branlard, and Nicholas Walker at DESY for providing us with insights into operating mobile robots at particle accelerator facilities.

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
