# OpenReview forum: "Towards Robust Exocentric Mobile Robot Tele-Operation in Mixed Reality"
_humanrobotinteraction.org/HRI/2022/Workshop/VAM-HRI — VAM-HRI 2022_

### Official Review · Reviewer_aKUq · 2022-02-25
**Interesting and relevant paper to VAM-HRI, accept**

**Rating:** 8
**Confidence:** 5

**Review:**

This paper presents an interesting method for operating remote robots with Mixed Reality. The overall approach is interesting, but clarification based on feedback below would help in clarifying the advantage of MR for this setting over VR.

1. The first two sentences of the paragraph are very awkward and seem grammatically incorrect and should be rewritten.
2. In Section II.C, remove “could” from first sentence
3. Future studies on this system should also evaluate quantitative task metrics such as task completion time and success rate.
4. For the real-world use case, is there a difference between using an MR and VR setting here? The final paragraph of Section 4 uses VR instead of MR which is confusing since it hasn’t been mentioned before, and the real-world use case makes it seem like there would be no use of the MR setup over a complete VR setup.

---

### Official Review · Reviewer_aNjs · 2022-03-01
**Exocentric MR teleoperation, accept**

**Rating:** 7
**Confidence:** 5

**Review:**

This paper describes a system that allows a user to remotely teleoperate a mobile robot using gestures in mixed reality. Two different modalities of teleoperation are proposed: (1) setting waypoints with a hand or controller and (2) free-hand trajectory drawing. A specific application is detailed, consisting of using a robot to perform maintenance inside a particle accelerator. The authors suggest that future work will include evaluating their design with validated assessments such as the SUS and TLX. This is an interesting and relevant paper; please see some questions below that will hopefully help fuel discussion at the workshop.

- Can the authors clarify how exocentric images would be projected from an egocentric camera on-board a remote mobile robot? This becomes clearer at the very end of the paper, but could use some clarification up front.
- Can you elaborate on the novelty of the path-following algorithm?
- In Section III, it states, "However, a particle accelerator is a large-scale facility, using the conventional egocentric MR tele-presence system could not facilitate task planning that involves large space and long-distance." Can you expand on why egocentric MR is not appropriate for this purpose?

Minor edit:
- In Section II A, ROS is for "Robot Operating System"

---

### Decision · Program_Chairs · 2022-03-04

Accept